# The Energy Transfer Yield between Carotenoids and Chlorophylls in Peridinin Chlorophyll *a* Protein Is Robust against Mutations

**DOI:** 10.3390/ijms23095067

**Published:** 2022-05-03

**Authors:** Francesco Tumbarello, Giampaolo Marcolin, Elisa Fresch, Eckhard Hofmann, Donatella Carbonera, Elisabetta Collini

**Affiliations:** 1Department of Chemical Sciences, University of Padova, Via F. Marzolo 1, 35131 Padova, Italy; francesco.tumbarello.1@studenti.unipd.it (F.T.); giampaolo.marcolin@studenti.unipd.it (G.M.); elisa.fresch@studenti.unipd.it (E.F.); donatella.carbonera@unipd.it (D.C.); 2Faculty of Biology and Biotechnology, Ruhr-University Bochum, D-44780 Bochum, Germany; eckhard.hofmann@ruhr-uni-bochum.de

**Keywords:** photosynthesis, light harvesting, energy transfer, carotenoid, peridinin, chlorophyll, PCP, N89L, two-dimensional electronic spectroscopy, 2DES

## Abstract

The energy transfer (ET) from carotenoids (Cars) to chlorophylls (Chls) in photosynthetic complexes occurs with almost unitary efficiency thanks to the synergistic action of multiple finely tuned channels whose photophysics and dynamics are not fully elucidated yet. We investigated the energy flow from the Car peridinin (Per) to Chl *a* in the peridinin chlorophyll *a* protein (PCP) from marine algae *Amphidinium carterae* by using two-dimensional electronic spectroscopy (2DES) with a 10 fs temporal resolution. Recently debated hypotheses regarding the S_2_-to-S_1_ relaxation of the Car via a conical intersection and the involvement of possible intermediate states in the ET were examined. The comparison with an N89L mutant carrying the Per donor in a lower-polarity environment helped us unveil relevant details on the mechanisms through which excitation was transferred: the ET yield was conserved even when a mutation perturbed the optimization of the system thanks to the coexistence of multiple channels exploited during the process.

## 1. Introduction

Cars are pigments that are ubiquitously present in the light-harvesting (LH) apparatus of photosynthetic organisms, where they usually play the role of “accessory” pigments that ensure photoprotection and enhance the absorption of LH complexes in the green region of the solar spectrum [1]. In order for the energy absorbed by Cars to reach the reaction center, a prior transfer to the nearby Chls must occur; Car-to-Chl ET thus represents a fundamental step in the early events of photosynthesis, and yet many aspects of its mechanism remain unclear, mainly because of the dark character of the lowest excited electronic state, or states, of the Cars.

Due to the C_2h_ symmetry of the conjugated chain, the photophysics of all-*trans*-Cars is classically interpreted in terms of three states, namely, S_0_, S_1_, and S_2_, with symmetries 1^1^A_g_^−^, 2^1^A_g_^−^, and 1^1^B_u_^+^, respectively [2,3,4]. In this picture, the excitation of the S_2_ state is strongly allowed, while the S_1_ state is symmetry-forbidden and can only be populated via rapid (~100 fs [5]) internal conversion. However, these simple symmetry considerations do not account for the whole spectral complexity of Cars. First, the strong coupling of electronic and nuclear motions and the presence of possible asymmetric substitutions break the symmetry selection rules [6]. Moreover, some photosynthetic complexes bind peculiar Cars, such as Per, shown in Figure 1a, in which an electron-withdrawing carbonyl group conjugated with the π-electron system stabilizes an intramolecular charge transfer (ICT) state [7,8,9,10]. The relationship between ICT and the other electronic states was extensively discussed in the literature [11,12,13,14,15,16,17], and it was often argued that an ICT character is associated with the dark S_1_ state. Such an “S_1_/ICT” state is believed to have an enhanced capability to act as an energy donor in photosynthetic ETs [18,19,20]. The picture was made even more intricate by the detection of further intermediate states between S_2_ and S_1_ [21,22,23,24,25], the most debated of which being the so-called “S_x_” state. Some works interpret S_x_ as an additional dark electronic state [21,26,27], while others describe S_x_ as a distorted conformer on the S_2_ potential energy surface, located in a local minimum [28] or near a transition state barrier between a planar and a twisted geometry [29,30,31,32]. Whether S_x_ also exists in free Cars in solution or only in Cars embedded in LH complexes, these works seem to converge on the interpretation of S_x_ as a potential ET channel to the Chls.

Intending to provide new pieces of evidence for the comprehension of Car-to-Chl ET, we focused our attention on the PCP, a soluble antenna of dinoflagellate algae, which developed a unique strategy to maximize the efficiency of underwater photosynthesis [20,33]. The wild-type (WT) form of PCP from *Amphidinium carterae* (Figure 1b) is a heterodimer that binds a cluster of Per and Chl *a* molecules in an unusual 4:1 ratio [34] and aggregates into a trimeric quaternary structure. By exploiting the strong S_2_ absorption of Per, PCP captures the green light that survives the overlying water column and then transfers energy to the Chl *a* with an ~95% efficiency [35]. Per is thus the principal absorber in PCP, and the energy flow from Per to Chl *a* is the crucial step in the function of this complex. Early transient absorption studies on PCP with an ~100 fs time resolution showed that most of the energy is transferred to the Q_y_ state of the Chl *a* in 2–3 ps [18,19,20,35] via a Forster-like ET from the S_1_/ICT state after internal conversion from S_2_ has occurred. The same studies also demonstrated that a significant portion (~25%) of the energy is delivered directly by the S_2_ state of Per [36] before the internal conversion, possibly to the Q_x_ state of Chl *a*. More recently, 2DES [37,38,39] measurements have unveiled new details on the direct ET pathway from S_2_, generating a debate on a possible coherent contribution to the ET [28,40,41,42,43].

The relative weights of the different ET channels, the factors affecting their efficiencies, and the involvement of further intermediate states of the Car in the ET are all open questions on the workflow of this one-of-a-kind antenna. A promising strategy to survey the photophysics of the PCP complex is to compare its WT form with refolded mutant complexes and assess whether and how the modification of specific structural and energetic features perturbs the energy flow between the pigments.

In the present work, we carried out a comparative analysis of WT PCP and the refolded N89L mutant, in which asparagine-89, a residue close to the conjugated chain of Per-614, is replaced with leucine. The N89L mutant is a homodimer reconstituted from the N-terminal half of the PCP polypeptide. The comparison between the crystallographic structures of the N89L protein and the refolded PCP homodimer (RF PCP), which is the basis for the N89L variant, showed that the mutation does not affect the structure of the complex [44,45]. Given the structural and spectroscopic equivalence between WT and RF PCPs [41,46], the WT PCP itself can be used for a meaningful comparison with the N89L mutant.

Per-614 is believed to be the Per of the cluster with the reddest energy site and the one showing the strongest interaction with Chl *a* [44]. The mutation shifts the red tail of the S_2_ absorption band toward higher frequencies. This can be clearly seen by comparing the absorption spectra in Figure 1c. A significant part of the oscillator strength of the band above 17,000 cm^−1^, which is associated with the S_2_ transition, is blue-shifted in the mutant protein. Magnetic spectroscopies already demonstrated that the photoprotective function of Per-614 is preserved in the mutant despite the different energy sites of this pigment [46].

Here, we investigated the excitation energy flow in the WT and N89L proteins in the ultrafast (<1 ps) time regime using 2DES. Thanks to a time resolution of about 10 fs and the inherent multidimensionality, 2DES overcomes most of the difficulties connected to the short time scales of photosynthetic ETs and the spectral congestion that is typical of multichromophoric systems. In a 2DES experiment, a sequence of three ultrashort laser pulses is focused on the sample and triggers the emission of a signal electric field. The data are typically visualized as a sequence of two-dimensional spectra acquired for different values of the time delay between the second and the third pulse (the “population time”, often indicated as t_2_). In each 2D map, the “excitation frequency” identifies the state excited by the first pulse, while the “detection frequency” provides information about the state probed by the third pulse. Therefore, off-diagonal peaks offer a powerful tool to detect phenomena coupling different states, such as internal conversion and energy transfer processes, and can help to characterize dark electronic states through their couplings with bright states [25,26,47,48]. More details about the experimental setup and sample preparation can be found in Section 3.

This work presents the first comparison of the WT PCP with its N89L mutant at the level of detail and with the temporal resolution offered by 2DES. Our approach demonstrated that the study of specific mutations can be effectively used to extract crucial information on the photophysics of protein-bound Cars. Moreover, this comparison sheds new light on the Car-to-Chl ET mechanism; as will be extensively discussed in the next sections, the cooperation of multiple ET channels may be a key factor in ensuring the well-known, but hardly understood, efficiency of this process.

## 2. Results and Discussion

The laser spectrum used for 2DES measurements, shown in Figure 1c, was tuned to cover (i) the Q_y_ transition of Chl *a* (670 nm, 15,000 cm^−1^); (ii) the band at 620 nm (16,100 cm^−1^), which includes the contributions from the Q_x_ transition of the Chl *a* and a vibronic Q_y_’ transition; and (iii) the red tail of the S_2_ transition of the carotenoid (<580 nm, >17,000 cm^−1^). The chosen bandwidth allowed us to capture possible signatures of coupling between Per and Chl *a* directly as cross-peaks in the 2DES maps. Moreover, this bandwidth acts as a spectral filter to specifically select the photophysics of the reddest Per within the cluster. We were thus able to selectively study the energy flow from the reddest Per to the Chl *a* and exclude from the analysis the processes through which the excitation descended from the higher energy Pers to the lowest energy one. In this way, it was possible to focus our attention on the spectral region that is most affected by the mutation.

In Schulte et al. [44], the major contribution to the shift of the S_2_ absorption band toward higher energies in N89L was associated with a destabilization of the two lowest-lying excited states of Per-614 as a result of the lower-polarity environment around this pigment in the mutant. It was also suggested that this blue shift could be such that Per-614 is no longer the reddest in the cluster [44]. However, the attribution of the reddest S_2_ transition to Per-614 provides only a simplified interpretation of the spectroscopic data: a more accurate description of the electronic structure of the cluster requires considering the formation of delocalized excitonic states [49], with the lowest energy one being localized mostly on Per-614 [43,50]. In any case, it is worth highlighting that the interpretation of 2DES data is not affected by this attribution since the final aim was to provide information on the relaxation dynamics following photoexcitation of the reddest states of the Per cluster manifold.

Absorptive 2DES maps at selected values of the population time t_2_ recorded at room temperature (295 K) are shown in Figure 2.

The diagonal signal centered at 15,100 cm^−1^ and the cross-peak at (16,100, 15,100) cm^−1^ are the typical spectral signatures of the Chl *a*, as already discussed in previous work [42]. The former is assigned to the ground state bleaching (GSB) and stimulated emission (SE) of the Q_y_ transition, while the latter is due to an internal conversion from the Q_x_ to the Q_y_ state, as well as a coupling of the Q_y_ electronic transition with vibrational modes at ~1000 cm^−1^ [42,51,52]. In the following paragraphs, attention will be focused on the signals appearing at the excitation frequency of Per (17,066 cm^−1^), as these contain the information about the processes through which the excitation migrates from the S_2_ state initially prepared down to the lower-lying states of Car, and eventually, to the Q bands of Chl *a* (Figure 3a). Two signals appearing at this excitation frequency were of particular interest. First, the broad negative bands at the detection frequency > 15,500 cm^−1^ (pinpointed with the square and the circle markers in Figure 2), were attributed to excited-state absorption (ESA) signals of Per. Second, the positive cross-peak at detection frequency of the Q_y_ state of Chl *a* (15,100 cm^−1^, triangle), which was direct evidence of the ET from Per to the Q_y_ state of Chl *a*. When qualitatively comparing the spectra of the WT protein to those of the N89L mutant, no appreciable discrepancies could be observed in the positions of the signals. We then looked for possible differences in the dynamic behavior of the signals.

As mentioned above, the 2DES experiment provides a stack of 2D spectra for different values of the time interval t_2_ between the second and the third pulse. By analyzing the evolution of the signal at different coordinates as a function of t_2,_ it is possible to gain valuable information on the various pathways that regulate the relaxation dynamics of the system. This evolution typically includes both non-oscillating exponential decay contributions (“populations” dynamics) and beating components (“coherence” dynamics) [53,54]. The temporal traces shown in Figure 3b–d were obtained by sampling the 2DES map at different combinations of excitation and detection frequencies and plotting the signal as a function of t_2_. These traces show an intense beating behavior superimposed on the exponential trends: the analysis results presented below first show the population dynamics, while the oscillating components are addressed afterward.

First, we studied the time evolution of the signal at the excitation frequency of Per and detection frequency of the Q_y_ state of Chl *a* ((17,066, 15,100) cm^−1^, triangle in Figure 2). This signal can be associated with the ET from Per to the Q_y_ state of Chl *a*, as graphically represented by the Feynman diagrams reported in Appendix A. The intensity of the signal at these coordinates was increasing with increasing values of t_2_, and the time constant regulating the signal rise thus provided an estimate of the effective rate of the ET. In the first place, we used a parallel (multi)exponential model to fit the temporal traces of the signals without introducing any a priori choice of a kinetic scheme. The results of the parallel fitting are summarized in Table 1 (further details on the fitting procedures are reported in Appendix A). As shown in Figure 3b, single exponential rises correctly reproduced the non-oscillatory dynamics, with time constants of about 1.9 ps and 2 ps for the WT and the N89L protein, respectively. These values are in the lower limit of the range reported in the literature for the S_1_/ICT→Q_y_ ET [14,18,19,20,28,35,40,41,42]. Different from a previous study [42], the ultrafast, possibly coherent, ET pathway from S_2_ could not be captured because of the different exciting conditions used here. Interestingly, the buildup of the ET signal in the mutant essentially retraced to the one in the WT, implying that the mutation did not induce significant changes in the rate of the ET process, at least in the first picosecond.

Moving the attention to the negative ESA signals, they most likely enclose contributions from different excited states of the Car, which cannot be easily identified based only on the coordinates. Indeed, the excitation frequency of these signals is that of the S_2_ state initially prepared, but the ESA can take place from dark states reached after the ultrafast internal conversion from S_2_; moreover, the detection frequency depends on the energy of the arrival state, which is generally unknown. Therefore, the identification of the electronic states contributing to the ESA must be based on the dynamics of the signals. Figure 3a clearly shows that the intensity of the ESA band was distributed in two “lobes”: a lower detection frequency signal (indicated with a circle, “lower” ESA) peaking at (17,066, 15,850) cm^−1^ and a higher detection frequency signal (square, “higher” ESA) peaking at (17,066, 16,700) cm^−1^. As is extensively discussed below, the two ESA lobes originated from different electronic states in the manifold of Per, as they showed different temporal dynamics.

The higher ESA signal for both the WT and the mutant proteins appeared within a time window comparable with the temporal resolution of the experiment (tens of fs). Moreover, as highlighted in Figure 3c, this signal had a lower initial intensity for the mutant than for the WT. In agreement with previous evidence [28,42], we attributed this signal to the S_1_/ICT→ S_n_ ESA. Indeed, the immediate onset of this signal in both samples could be justified by invoking the presence of a conical intersection (CI) between the S_2_ and the S_1_ states [28]. The sub-10 fs population of the S_1_ state through the CI from the initially excited S_2_ explained why this negative signal was recorded immediately after excitation. The difference in the signal intensity at early times (Figure 3c) could instead be associated with a lower quantum yield across this CI for the mutant than for the WT.

To investigate the factors affecting the yield of the crossing between the two electronic surfaces, we focused our attention on the vibrational degrees of freedom that were closely involved in the dynamics of the CI. Theoretical models describing CIs distinguish between coupling modes, which induce electronic coupling, effectively forming the CI, and tuning modes, which tune the energy gap between the involved electronic states and thus regulate the access to the CI [55,56,57,58]. Paramount information on the vibrational modes active in the CI can be found in the so-called “power spectra”, shown in Figure 4, which reveal the main oscillating components contributing to the 2DES signal. They are obtained by Fourier-transforming the oscillating residues integrated over the two frequency dimensions [53]. The three most prominent beating components are all easily attributed to vibrational modes of Per; the contribution of the Chl to the power spectra is expected to be marginal due to the lower Huang–Rhys factor of its vibrational modes. The two vibrations at 1160 and 1225 cm^−1^ are attributed to the C-C stretching mode of Per mixed with the C-H in-plane bending mode, while the vibration at 1564 cm^−1^ is associated with the C=C stretching mode of the carotenoid [59,60,61]. It is noteworthy that the C=C stretching mode, already indicated as the tuning mode of the CI both in Cars [28,57] and retinal [56], showed a reduced amplitude in the N89L mutant with respect to the WT protein.

It was argued that the C=C/C-C bond order reversal of the conjugated polyene backbone that accompanies the displacement along the C-C and C=C vibrational coordinates provides an ICT character to Per [11]. Therefore, after optical excitation of the S_2_ state, the C=C stretching mode should push Per towards the CI in half a period of the vibration (~10 fs) while developing an ICT character. The lower quantum yield of the CI could, thus, be ascribed to a poorer stabilization of the ICT character assumed by Per at the intersection between S_2_ and S_1_ in the lower polarity environment of the mutant. Indeed, the polarity-dependent stability of the ICT state is a well-documented feature [8] and was recently observed by Marcolin et al. [12] in the 2DES spectra of fucoxanthin, a carbonyl Car similar to Per.

An alternative interpretation of the higher ESA signal was provided by Beck and co-workers [29,30,31,32,40,41,43], who assigned it to the absorption of a displaced S_2_ structure with a relatively long intrinsic lifetime labeled S_x_ [40,41,43]. It is worth noting that this assignation is not in contrast with the key arguments discussed above. Indeed, they describe S_x_ as an S_2_ conformer with an ICT character, which is formed after the motion along the bond-length alternation coordinate C=C/C-C and initial twisting. In their picture, however, S_x_ is formed immediately directly through the evolution in the S_2_ potential energy surface and the CI between the S_2_ and S_1_ surfaces is only reached later. It should however be noted that in those works, a bluer excitation profile was used, likely leading to the excitation of Per transitions that were different from the reddest transition addressed in our experimental conditions and, thus, to a slightly different dynamical evolution.

The second relevant difference between the WT and the mutant in the dynamics at the higher ESA coordinates was the time constants estimated for its decay, corresponding to the S_1_/ICT relaxation (Table 1). The values of these constants were affected by a high uncertainty because they were much longer than the time window investigated (700 fs). Nonetheless, it can be clearly seen that the N89L mutant showed a significantly slower decay. Since it is known that the de-excitation of the S_1_/ICT state is mainly promoted by transfer to the Chl in the ps temporal regime [18,19,20,35], these data indicated a slower transfer to the Q_y_ state of Chl *a* for the mutant. This could be reasonably ascribed to the poorer overlap between the S_1_/ICT and the Q_y_ band caused by the destabilization of the S_1_/ICT state in the lower-polarity environment of the mutant. It is significant that because the rise of the ET signal is faster than the decay of the S_1_/ICT ESA, then the S_1_/ICT → Q_y_ transfer could not be the only channel exploited in PCP for the overall Per-to-Chl *a* ET.

When compared to the higher ESA, the temporal evolution of the lower ESA signal exhibited a different trend. It can be more properly fitted by using an exponential rise (115 fs) followed by a decay (630 fs). This indicated that the state from which the lower ESA originates was a different excited state in the manifold of Per. As mentioned in the introduction, intermediate Car states with different features (and labels) were reviewed in the literature [21,22,23,24,25,26,27,28,32]. We labeled the state generating the lower ESA as S’ to remain general and postpone the definitive identification of this state to a future, more targeted study. The notable intensity of the lower ESA at early times in Figure 3d suggested that S’ could have been directly excited by the laser pulse. A possible assignation of S’ would be a distorted S_2_-like state excited via transitions from twisted ground-state conformations, presumably favored by the protein scaffold and expected to lie in the red-edge of the Per absorption [29,62,63].

Moreover, the dynamic behavior in Figure 3d indicated that an additional population could be brought into S’ within a few hundreds of fs, possibly via a torsional motion from the “proper” (planar, all-*trans*) S_2_ state. Indeed, the population of twisted carotenoid structures through torsional distortion of Per in the S_2_ state is a hypothesis that is commonly reviewed in the literature [21,22,29,30,31,32]. In support of this hypothesis, we noticed that the peak at ~50 cm^−1^, which is recognizable in the power spectra of Figure 4, could be attributed to a low-frequency torsional mode. A vibrational coherence peaking at a similar frequency was detected in retinal, which has an analogous polyene backbone [56,64]. Such a torsional mode could guide Per to the S’ state within half a period of oscillation (~350 fs), passing through a series of intermediate distorted structures. Our experimental observations on S’ agreed with several already recognized properties of the S_x_ state, above all, its capability to directly transfer energy to Chl [29,62,63]. This evidence points toward the assignation of the decay of the lower ESA to the S’→Q_y_ ET process. In addition, as summarized in Table 1, the fitting of the lower ESAs highlighted a higher amplitude of the rising component for the N89L mutant. This was compatible with a larger S_2_ population that survived the CI and underwent torsional distortion to the S’ state. The higher intensity of the low-frequency torsional mode in the power spectrum of the mutant (Figure 4) also seemed to correlate with the greater population that reached the distorted S’ state.

We want to highlight that an alternative interpretation of S’ as a vibrationally “hot” S_1_/ICT state cannot be excluded. A hot S_1_/ICT state could be directly excited by borrowing oscillator strength from the almost-resonant S_2_ transition and further populated by internal conversion. Moreover, the involvement of a vibrationally excited S_1_/ICT state as an additional channel for the ET process was already proposed [18].

Altogether, these findings suggested the presence of at least two synergic pathways contributing to the overall ET from Per to Chl *a* in the investigated time and frequency range. Indeed, we could exclude the idea that in these conditions, the only channel was the S_1_/ICT→Q_y_ transfer because the rise of the ET signal (triangle, Figure 3b) was faster than the decay of the S_1_/ICT ESA (square, Figure 3c). Moreover, the overall ET rate, estimated as 1.9 and 2 ps for the WT and the mutant, respectively (Table 1), remained practically unchanged in the two samples, despite the slower decay rate of S_1_/ICT in the N89L mutant. We thus proposed a kinetic model like the one represented in Figure 5a in which the S_1_/ICT and S’ states cooperate to transfer energy to the Q_y_ state of Chl *a*. Based on this kinetic scheme, it was possible to devise a new fitting model (Appendix A) that allowed for retrieving the kinetic constants associated to each of the identified transfer pathways and their relative weights for the WT and the mutant, as illustrated in Figure 5b. The results obtained after applying this kinetic model clearly outlined that the mutation did not alter the kinetic constants relevant for the ET; instead, the ET rate was preserved thanks to a redistribution of the relative weights of the two ET channels.

Indeed, according to the model in Figure 5a, once the reddest Per was excited in the S_2_ state, part of the S_2_ population was transferred to the S_1_ state through a CI in a timescale comparable with the temporal resolution of the experiment. The CI exhibited a lower yield in the mutant, possibly because of polarity or steric effects affecting the vibrational mode that regulated access to it. The excitation energy was then transferred from S_1_ to the Q_y_ state of Chl *a* in a few ps ($k_{S_{1}Q_{y}}^{-1}\sim$3 ps in the WT protein, ~8 ps in the mutant). In parallel, the population in S_2_ was pushed via torsional motions through a series of distorted structures. Eventually, this led to the rise of a state labeled S’ on a time scale of $k_{S^{\prime },R}^{-1}$ = 409 fs (395 fs in the N89L mutant). From S’, the excitation energy could be efficiently transferred to the Chl *a*, with a calculated time constant of $k_{S^{\prime }Q_{y}}^{-1}$ = 501 (503) fs for the WT (N89L) sample. Altogether, during the first 700 fs, in the WT, ~85% of the excitation was transferred to the Chl *a* starting from the S’ state, while the remaining 15% came from the S_1_ state, as shown in Figure 5b; the contribution from S’ was more significant in the mutant (~94%) due to the larger population that survived in S_2_ and moved to S’. Overall, the ET process in the N89L mutant suffered only a slight slowdown thanks to the compensating effect of the S’ channel.

## 3. Materials and Methods

In a 2DES experiment, the sample is excited by a sequence of three ultrashort laser pulses (with wavevectors **k_1_**, **k_2_**, and **k_3_**), separated by time delays t_1_ (between **k_1_** and **k_2_**) and t_2_ (between **k_2_** and **k_3_**). The signal emitted by the sample is then detected as a function of a third time interval t_3_ (between **k_3_** and the signal).

In our experimental setup, fully described in [65], the pulse sequence is generated starting from the output of an amplified pulsed laser source (Libra Coherent, Santa Clara, CA, US), which produced a train of 100 fs pulses centered at 800 nm with a repetition rate of 3 kHz. The central wavelength of the pulse was then tuned in the Vis range by using a non-collinear optical parametric amplifier (TOPAS White Light Conversion, Vilnius, Lithuania). The pulse was then compressed and shaped with a prism compressor and an acousto-optic programmable dispersive filter (Dazzler Fastlite, Antibes, France). The resulting pulse was 100 nm broad and centered at 620 nm; the pulse duration at the sample position was ~10 fs, as measured through the frequency-resolved optical gating (FROG) technique (Appendix A). The pulse energy was attenuated to 12 nJ using a broadband half-waveplate/polarizer system. A diffractive optic element split the pulse into four identical replicas: three replicas served as the exciting pulses **k_1_**, **k_2_**, and **k_3_**, while the fourth was attenuated with a graduated neutral filter and used as a local oscillator (LO) for detection purposes (vide infra). A donut-shaped spherical mirror (DSM) arranged the exciting pulses along the directions **k_A_**, **k_B_**, and **k_C_**, representing three vertices of an ideal square, and the LO along the fourth vertex (BOXCARS geometry). The time delays between pulses were controlled by means of couples of antiparallel CaF_2_ wedges. One wedge of each pair was mounted onto a translation stage (Ant95 Aerotech, Pittsburgh, PA, US) that regulated the thickness of the medium crossed by the exciting beam and provided a temporal resolution of 0.07 fs. After a second DSM focused the four beams on the sample, the latter emitted a third-order signal in the phase-matched direction **k_S_** = −**k_A_** + **k_B_** + **k_C_**. This direction coincided with the fourth vertex of the BOXCARS square and, thus, with the direction of the LO. The signal was detected as an interference with the LO (heterodyne detection) via a scientific complementary metal–oxide semiconductor (sCMOS) camera (Zyla Andor, Oxford Instruments, Belfast, Northern Ireland) after a spectrograph (Shamrock 303i Andor) had separated its frequency components. The setup also included two optical choppers to remove spurious contributions from the signal via the double lock-in modulation method [66].

The experiment is conducted by scanning the time delays t_1_ (from −80 to 100 fs in 1 fs steps) and t_2_ (from 0 to 1 ps in 5 fs time steps). The acquired data matrix was thus a function of t_1_, t_2_, and ω_3_, where ω_3_ was the frequency dimension read directly by the sCMOS camera, corresponding to the Fourier transform of the third time interval t_3_. By controlling the relative arrival time of the pulses, two types of experiments were performed: in the rephasing (R) experiments, the pulse propagating along the direction **k_A_** arrived first (**k_A_** = **k_1_**, **k_C_** = **k_2_**, and **k_B_** = **k_3_**, thus **k_S_** = −**k_1_** + **k_2_** + **k_3_**); in the non-rephasing (NR) experiments, the pulse propagating along the direction **k_C_** arrived first (**k_C_** = **k_1_**, **k_A_** = **k_2_**, and **k_B_** = **k_3_**, thus **k_S_** = +**k_1_** − **k_2_** + **k_3_**). The experiment on each sample was repeated five times to ensure reproducibility and averaged to reduce noise. After a processing procedure that included a Fourier transform along the t_1_ axis, the data were arranged as a stack of two-dimensional (ω_1_, ω_3_) spectra at different t_2_ times. The analysis presented in this work was conducted on the total (T) purely absorptive data, which was obtained as the sum of the R and NR data matrices. Further details on the experimental setup and the calibration, acquisition, and data processing procedures can be found in Bolzonello et al. [65].

The sample of WT PCP was prepared as detailed in Meneghin et al. [42]. The N89L sample was prepared as described in Schulte et al. [44] and diluted in a buffer containing tricine 5 mM, KCl 2 mM, and NaCl 30 mM at pH = 6.5 (all reagents were supplied by Merck, Darmstadt, Germany) until an absorbance of about 0.4 in a 1 mm cuvette was achieved in the region of the Q_y_ band of Chl *a*. Such an optical density ensured a good compromise between the need for a third-order response of sufficient intensity and the avoidance of a drastic attenuation of the LO beam. In order to prevent the formation of harmful oxidizing species, oxygen was removed from the solution by degassing the sample under nitrogen flux and the cell was sealed. For each sample, steady-state absorption spectra were acquired before and after the 2DES experiments to verify that no degradation had occurred during the measurements.

## 4. Conclusions

The comparison of the ultrafast dynamic behavior of the WT PCP and a refolded N89L mutant allowed for unveiling important information about the workflow of this light-harvesting complex. While previous studies had already shown that the N89L mutation does not affect the ET efficiency [44], here we shed light on the mechanisms underlying such robustness. By exploiting the multidimensionality of the 2DES technique, two parallel channels for the ET could be identified: along with the well-documented pathway from the S_1_/ICT state, transferring excitation to the Q_y_ band of the Chl *a* in a few ps [18,19,20,35], we recognized the crucial role of a further intermediate state donating energy in the first hundreds of fs after photoexcitation. This second channel becomes more relevant in the mutant, where the S_1_/ICT channel was partially undermined because of the mutated energy landscape. These findings suggest that the cooperation of multiple pathways might be decisive in ensuring high ET performance even if a pathway is compromised to a certain degree. Although this conclusion was drawn from data collected in specific experimental conditions, recent evidence in the literature seems to indicate that in effect it might have more general validity.

First, other works have demonstrated the robustness of the ET in PCP against the replacement of Chl *a* with different Chls [43,67]. Second, the overview of the most recent 2DES works on PCP revealed that different exciting conditions might favor specific de-excitation pathways above others by preparing different initial states of the Per donor. The exciting conditions used in this work placed a magnifying glass on the photophysics of Per structures in the red edge of the spectrum, which can be reasonably associated with distorted carotenoid geometries, possibly designed by the protein environment. Their contributions to the spectroscopic signal would have been elusive with a bluer laser band. Indeed, with a more significant excitation of the S_2_ state, the direct S_2_→Q_y_ ET channel and the population of the S_1/_ICT state from S_2_ via CI would have become the dominant de-excitation pathways. The strong sensitivity of the spectroscopic response to the exciting conditions also justifies, for example, the multifarious interpretations proposed for the “S_x_” state and its role in the ET, and the number of different possible ET pathways so far identified [28,40,41,42,43]. Upcoming analyses with a greater focus on the amplitude and phase distribution of the Car vibrational modes over the 2DES map could provide more definitive evidence on the identity of the states detected in 2DES measurements.

Beyond implying that particular care must be paid in the comparison of literature data, this vast wealth of investigations is progressively unveiling the complexity and multiplicity of the mechanisms regulating the efficiency of ET in the PCP antennas. The photophysical properties of PCP have carefully evolved to be robust against mutations potentially threatening its light-harvesting function.

From a broader perspective, it is likely that the diversity and complementarity of channels available for the ET may be a common strategy affecting the robustness of photosynthesis on a biologically relevant scale. Future investigations on other natural antennas would help understand whether Car-to-Chl ET mechanisms similar to the one described in this work for PCP are also relevant in LH complexes different from PCP.

## Figures and Tables

**Figure 1 ijms-23-05067-f001:**
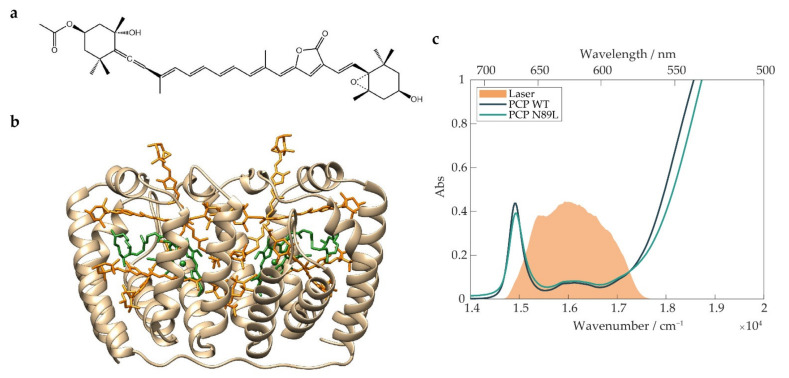
(**a**) Structure of the carotenoid Per. (**b**) Crystallographic structure of WT PCP, with the Per molecules colored in orange and the Chl *a* molecules colored in green. (**c**) Absorption spectra of WT PCP (blue) and its N89L mutant (green). The spectrum of the exciting pulses used in 2DES measurements is also shown (orange area).

**Figure 2 ijms-23-05067-f002:**
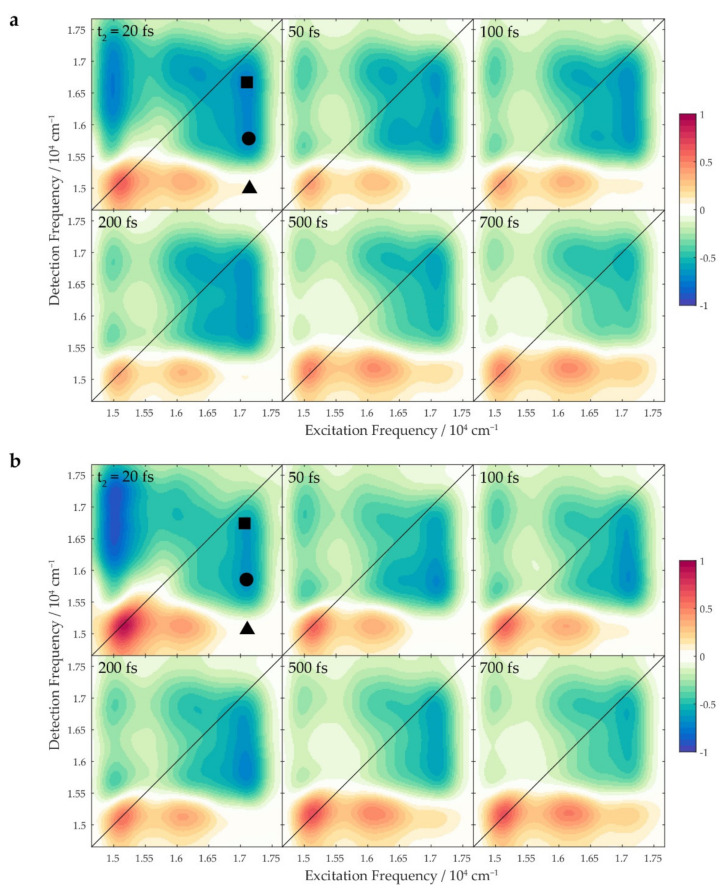
Absorptive 2DES maps of (**a**) WT PCP and (**b**) N89L PCP at different population times (t_2_); to make the evolution of the populations more evident, the oscillating contributions to the signal were attenuated using a Savitzky–Golay filter. The markers pinpoint relevant coordinates discussed in the main text.

**Figure 3 ijms-23-05067-f003:**
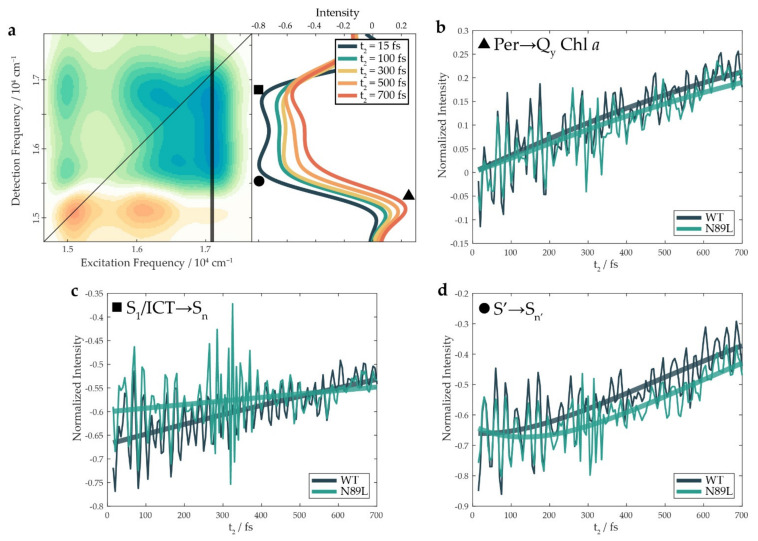
(**a**) Vertical cuts of the 2DES maps of WT PCP at excitation frequency 17,066 cm^−1^ (indicated with the black line) at different population times; to ease the visualization of the signal trends, the oscillating contributions to the signal were attenuated using a Savitzky–Golay filter. Temporal traces of (**b**) the ET signals (extracted at (17,066, 15,100) cm^−1^, triangle), (**c**) the higher frequency ESA signals (extracted at (17,066, 16,700) cm^−1^, square) and (**d**) the lower frequency ESA signals (extracted at (17,066, 15,850) cm^−1^, circle) for the WT and the mutant samples. Thick solid lines represent the fittings performed according to the multiexponential model described in Appendix A (only the non-oscillating components are shown for clarity).

**Figure 4 ijms-23-05067-f004:**
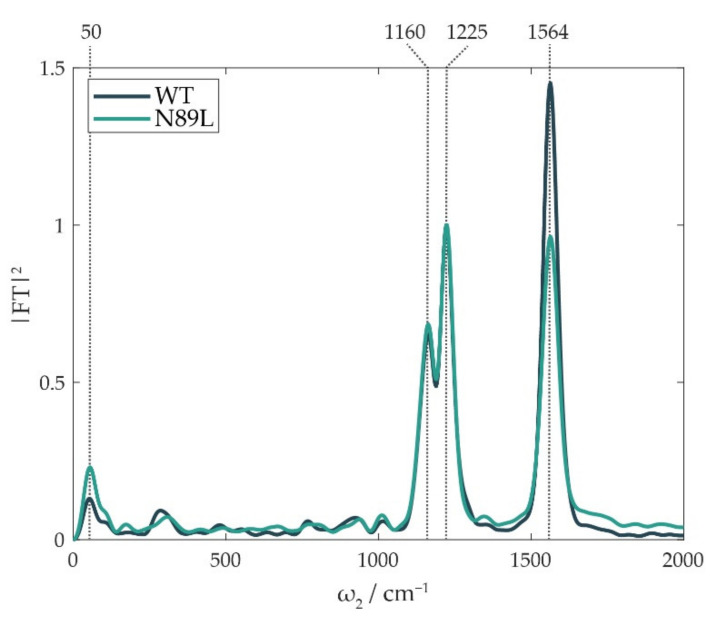
Fourier transform of the oscillating components of the 2DES signal, mediated along excitation and detection frequency axes. They are plotted as square moduli of the Fourier transform amplitudes and normalized on the intensities of the 1225 cm^−1^ component.

**Figure 5 ijms-23-05067-f005:**
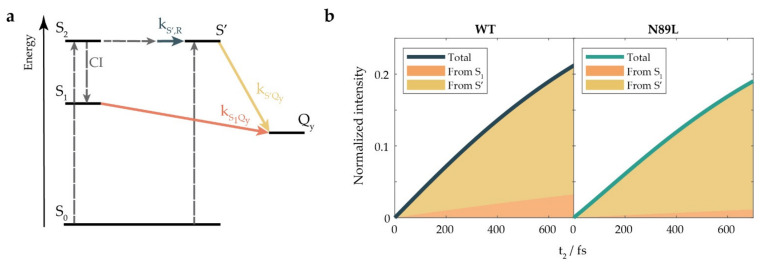
(**a**) Energy level diagram illustrating the different dynamic processes included in the kinetic model used to interpret the temporal dynamics of the experimental traces. Dashed lines indicate events occurring on a time scale faster or comparable with the time resolution of the experiment, namely, the initial optical excitation and the decay of the S_2_ state via (i) the CI that transferred the population from S_2_ to S_1_ and (ii) the initial torsional movements that promoted the formation of distorted structures. $k_{S^{\prime },R}$, $k_{S_{1}Q_{y}}$, and $k_{S^{\prime }Q_{y}}$ represent the kinetic constants for the rise of S’, the S_1_/ICT→Q_y_ ET, and the S’→Q_y_ ET, respectively. (**b**) Contributions to the rise of the ET signal, as determined by the fitting of the experimental data using the kinetic model in panel (**a**).

**Table 1 ijms-23-05067-t001:** Results of the parallel (multi)exponential models used to fit the traces at (17,066, 15,100) cm^−1^ (triangle, ET signal), (17,066, 16,700) cm^−1^ (higher ESA, square) and (17,066, 15,850) cm^−1^ (lower ESA, circle); positive (negative) amplitudes indicate rising (decaying) components.

	A1		T1/fs		A2		T2/fs	
	WT	N89L	WT	N89L	WT	N89L	WT	N89L
▲ ET signal	0.80	0.76	~1900	~2000				
∎ Higher ESA	−0.70	−0.62	~3000	~8000				
● Lower ESA	0.20	0.28	115	112	−0.67	−0.65	630	630

## Data Availability

All data supporting the findings of this study are available from the corresponding author upon request.

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
