# Peer review of "The Energy Transfer Yield between Carotenoids and Chlorophylls in Peridinin Chlorophyll a Protein Is Robust against Mutations"

_ijms, 2022, doi:10.3390/ijms23095067_

Round 1
Reviewer 1 Report
The authors describe an article entitled “The Energy Transfer Yield Between Carotenoids and Chlorophylls in PCP is Robust Against Mutations”. The topic of the manuscript is interesting, and the manuscript constitutes an interesting research article concerning photosynthesis.
The work is well-written and a well-constructed introduction has been established by the authors. Sufficient spectra and figures are included in the manuscript for comprehension and clarity. Interesting and convincing results are also presented in this work. Overall, I think that this is a manuscript that I recommend for publication after inclusion of minor revisions.
1) The conclusion can be improved. Indeed, at present, no perspective to this work is given by the authors.
2) From my viewpoint, no sufficient details are given in “Materials and Methods” in order the different experiments to be reproducible. Please improve.
3) the introduction section should be improved. In fact, the authors discuss from several family of compounds but no structure is given. Parallel to this, the authors discuss from “C2h symmetry of the conjugated chain in all-trans-Cars”. It would be convenient to present the different structures
Reviewer 2 Report
I find the poaper interesting and worth to be published. I would recommend some minor revisions;
Avoid using abbreviations in the title. Not all readers might be familiar with PCP.
The novelty of the paper is not clear. Please state clearly in the Introduction what was done in this work by the first time and why it is important. Authors say what they did in the work, but do not give an emphasis on what is novel, never done before, and that is very important.
Fig. 1 has too small letter size, it is hard to read.
Fig 4, the peaks should be identified in the graphic.
I do not understandFfig 5a. If it is temporal, where is the time axis?
